# Ancient Wheats—A Nutritional and Sensory Analysis Review

**DOI:** 10.3390/foods12122411

**Published:** 2023-06-19

**Authors:** Hala Roumia, Zoltán Kókai, Bernadett Mihály-Langó, Éva Csajbókné Csobod, Csilla Benedek

**Affiliations:** 1Department of Postharvest, Supply Chain, Commerce and Sensory Science, Institute of Food Science and Technology, Hungarian University of Agriculture and Life Sciences, Villányi Str. 29, 1118 Budapest, Hungary; halaroumia@gmail.com (H.R.); kokai.zoltan@uni-mate.hu (Z.K.); 2Cereal Research Non-Profit Ltd., Alsókikötő Str. 9, 6726 Szeged, Hungary; langob@gabonakutato.hu; 3Department of Dietetics and Nutrition Science, Faculty of Health Science, Semmelweis University, Vas Str. 17, 1088 Budapest, Hungary; csajbokne.csobod.eva@semmelweis.hu

**Keywords:** ancient wheat, underutilized, macronutrients, micronutrients, sensory analysis

## Abstract

The purpose of this review is to provide a critical evaluation of the nutritional and sensory properties of ancient wheats (spelt, emmer, einkorn, and kamut) and the methods used to analyze them. This paper provides a comprehensive overview of the main analytical methods applied to study the nutritional properties of ancient wheats. According to our findings, protein content was the most commonly studied macronutrient across all types of ancient wheat species. The article notes that einkorn bran showed the highest protein and ash content, which reveals the potential of ancient wheats to be more widely used in food products. Regarding the majority of amino acids in spelt wheat cultivars, the general trend in the data was rather consistent. This review also compares sensory evaluation methods for different wheat products made from ancient wheats, such as bread, pasta, cooked grains, porridge, snacks, and muffins. The various reported methods and panel sizes used prove that ancient wheat products have many potential sensory advantages. Overall, using ancient wheats in wheat products can enhance the nutritional benefits, increase diversity in the food systems, and may be more appealing to consumers looking for something different, thereby contributing to the development of more sustainable and locally based food systems.

## 1. Introduction

Wheat is one of the most significant staple foods worldwide and is present in significant quantities in a wide range of food products. Over 783 million metric tons of wheat were consumed worldwide in the marketing year 2020–2021, a slight increase from the previous year [1].

Despite the rising demand for high-quality food products, yield continues to be dominant and the primary determinant of profitability in the production of cereals. Advancements in breeding techniques are necessary for maximizing production. Low amounts of protein, some minerals (especially Fe and Zn), and other nutraceuticals are common characteristics of high-yielding crop types, which are also known to have poor nutritional quality [2,3]. As a result, common cereals are becoming less appropriate to produce nutritious foods. The Green Revolution offered only a temporary way to solve the world’s hunger problem but was not effective in the long term since the growth in food production was not followed by an increase in food’s nutritional value. Dysfunctional food systems that are unable to sustainably provide all necessary nutrients and health-promoting ingredients needed for human life are the root cause of malnutrition. Cereal food products are the main source of essential elements that are necessary for human health [4]. In certain industrialized and developing nations, there has been a gradual transition from more intensive cropping practices to more sustainable, low-input ones due to public health and environmental concerns. The present high-yield and high-protein wheat cultivars do have one disadvantage, though, in that they demand a lot of agricultural inputs, such as nitrogen fertilizers, insecticides, and herbicides. In order to change the current pattern of high input/low nutritional food quality, ancient wheats that have never been subjected to breeding programs and their landraces/original forms are currently available and may provide an important component in the development of low input/high nutritional food quality systems [5].

The major types of ancient wheat species are spelt, einkorn, emmer, and kamut [6]. By using the available ancient wheat species in our food products, it could be possible to reach a holistic and sustainable method to increase the biodiversity of cultivated cereals and also enrich our food diversity through cereals with valuable nutrients [7], where the ancient wheat genotype groups stood out as having significant mineral content, especially the highest levels of Zn and Fe [8]. For example, spelt wheat (*Triticum spelta*) is one of the ancient wheat species that may contain higher levels of protein, soluble dietary fiber, and minerals than those of common bread wheat (*Triticum aestivum*) genotypes [9]. The crude fat content of einkorn, emmer, and spelt was found to be significantly higher (*p* < 0.05) when compared to whole wheat flour made from hard red spring wheat [10].

Bread wheat (*Triticum aestivum)* has three sets of chromosomes (AABBDD) and is an allohexaploid [11], spelt wheat (*Triticum spelta)* is hexaploid [12], and both emmer wheat (*Triticum turgidum)* and kamut wheat (*Triticum turanicum)* are tetraploid [13,14]. Durum wheat (*Triticum turgidum*) is also a tetraploid species [15], and einkorn wheat (*Triticum monococcum)* is a diploid [16]. A recent study proved a strong consumer interest in ancient cereals, as nearly all participants expressed their willingness to purchase bread or other products made from ancient cereals. They also demonstrated a high level of awareness regarding different varieties of ancient cereals, with over 95% recognizing spelt, which can be attributed to its long history of use as a staple cereal. Consumers also showed high sensory acceptance of spelt. This increasing interest in ancient cereals is observed among consumers, bakeries, and farmers, with a focus on local crop production and sustainability trends [17].

## 2. Research Objectives

The research objectives of this study are to comprehensively review the methods applied to the analysis of ancient wheats, analyze the nutritional composition of ancient wheats, and review their macro- and micronutrient profiles, as well as their amino acid content, using a literature review of relevant research articles. Additionally, the study aims to evaluate the sensory characteristics of ancient wheats, such as taste, aroma, and overall acceptability. Furthermore, the study intends to compare the nutritional and sensory attributes of ancient wheats to modern wheat (*T. aestivum*) and identify potential health benefits associated with their consumption in various food products to enhance nutritional value and sensory appeal.

## 3. Materials and Methods

We conducted a literature review of the scientific literature and relevant research articles on ancient wheats, focusing on their nutritional composition and sensory properties. The literature search includes both traditional and online databases, such as Web of Science, Scopus, and Google Scholar, using keywords related to ancient wheat varieties.

## 4. Data Collection and Analysis

Nutritional Analysis: We collected and analyzed nutritional data and methods applied from the identified research articles, including information on macro- and micronutrients and amino acids of different ancient wheat varieties (spelt, einkorn, emmer, and kamut).

Comparative Analysis: We compared the methods applied to measure the nutrient content and also compared the nutritional profiles of ancient wheats with those of modern wheat varieties and analyzed the data to determine the differences and similarities between the two groups.

Sensory Analysis: We identified studies that have evaluated the sensory attributes of ancient wheats and analyzed the methods applied in sensory tests and the results to understand the sensory characteristics of different ancient wheat varieties.

## 5. Results

Nutritional Properties: The analysis of ancient wheats, including spelt, emmer, einkorn, and kamut, revealed distinctive nutritional properties compared to common bread wheat. These ancient wheats were found to have higher levels of certain nutrients, especially protein, fat, fiber, and minerals. Incorporating ancient wheat varieties into food products can provide additional nutritional benefits to consumers.

Sensory Evaluation: Understanding the sensory aspects of ancient wheat varieties is crucial for creating products that meet consumer preferences and expectations. The sensory evaluation of different types of ancient wheat products demonstrated their unique and pleasant sensory attributes.

Importance of Analytical Methods: Comparing measured parameters and methods applied enhances the organization, comparability, and synthesis of data, contributing to a better understanding of the research landscape and facilitating evidence-based decision-making in the field of nutrition and product development.

## 6. Origin and Definition of Ancient Wheats

In the mid-19th century, a group of botanists made the initial endeavor to identify the origin of cultivated wheat varieties. Notably, Linnaeus (1753) was the first to classify all cultivated wheats under the genus *Triticum*. Over time, the discovery of newly cultivated species and a deeper understanding of taxonomical relationships among different taxa have led to modifications in the classification of wheat varieties. However, since the wild ancestor of most cultivated wheats, such as emmer, durum, spelt, and common wheat, remained unknown, many scientists at the end of the 19th century believed that the original prototype had become extinct, leaving the exact origins of these wheats forever untraceable [18].

Ancient wheat varieties lack a common, clear, and precise definition; however, it is agreed that ancient wheat species have not changed throughout the past century, and the most popular ancient wheat species include spelt, einkorn, emmer, and khorasan. Research revealed that ancient wheats had a more valuable nutritional profile than modern common wheats [19].

The majority of ancient wheat has a glume that needs to be removed in the mill, giving it the name “hulled wheat”. Hulled wheat species have glumes (husks) that cover the wheat grains; the glumes are strong, hard, and resistant after harvest. The possible chaff contamination of harvested grain prevents its direct use in the food processing industry. Before processing, hulled wheat needs to have its glumes removed. A hulled wheat spike separates into spikelets during threshing. Dehullers are used to remove the hulls from the grain. Therefore, ancient wheats have the disadvantage that the hulls remain attached after threshing, making it difficult to thresh their grains. This means that additional processing is needed to remove the hulls or husks and prepare the grains for milling or pounding. Spikes need to be further processed after being harvested, a procedure known as threshing [20]. In the primitive wheat *Triticum spelta*, the rachis is brittle and the glumes are tightly attached [21]. However, a study was conducted in Poland on ancient wheat species, including einkorn, emmer, spelt, and bread wheat, to evaluate the presence of a selected pathogen (*Fusarium* spp.) in their grain and glumes, and the study revealed that the fungi pathogen is not able to penetrate the grain tissues. The results were promising, especially for einkorn glumes and grains, which could possibly be used to make bakery products that are hypoallergenic [22].

## 7. The Revival of Interest in Ancient Wheat Varieties

Durum and bread wheat varieties have excellent yields that make them widely grown worldwide; thus, they have mostly replaced spelt, einkorn, and emmer wheats. However, recently, there has been a growing interest in these neglected crops because hulled wheat species have advantages over bread wheat and durum wheat species in terms of nutritional qualities, diversification, and sustainability in agriculture [23].

Ancient wheats have been neglected for a long time in Europe; however, they are becoming more popular nowadays because they can sustainably provide a valuable nutritional profile. Ancient wheat varieties have been rediscovered by consumers, bakers, millers, and farmers. Spelt wheat (*Triticum spelta*) is one of the commercially accessible ancient wheat varieties. In the last decade, interest in ancient wheat species has been revived by the need for traditional food products to conserve genetic diversity and the demand for species that may be farmed in disadvantaged places with high adaptability [24].

## 8. Ancient Wheat Production and Yielding

Ancient grains are generally more suited to climate change and sustainable production methods than commercial grains, and this alone makes them essential as global food security crops that should be protected and widely planted. However, creating a strong and long-lasting demand is the best method to maintain investment in production technologies. The only way to guarantee customer demand is to provide evidence of the health advantages that these grains offer compared to commodity grains. The most significant restriction on the use of ancient grains is their low production. Their lower yield productivity compared to the common grains is a main factor in their low production. Perhaps even more concerning is the overall widening of the yield gap between the common grains and the ancient grains. Another important factor is the lack of significant involvement in ancient grains by the major international seed-breeding companies, which is a big barrier to increasing the productivity of ancient grains. Probably the issue here is simply that the market for the seeds is too tiny to make the expense of development worthwhile. It is also challenging to find evidence on breeding expenses versus financial benefits [25].

Although ancient wheats are underutilized plant species that are crucial for food security but are neglected in industrial production, they are growing in popularity due to their low-input organic cultivation and the superior nutritional content of their flour compared to modern wheat. The research findings on the processing of hulled wheat are promising and indicate that it is possible to produce a broader range of specialty goods; however, hulled wheat would not be suitable for the mass market due to low yields [20]. Ancient wheats grow tall, so they are vulnerable to lodging, which results in a large yield loss. On the other hand, from an economic perspective, ancient wheats are low-input crops that require less fertilizer and could be desirable crops in low-income countries [24].

While ancient grains can be grown with low inputs and are mostly cultivated in developing countries using conventional and organic-type agriculture, they cannot be easily produced in the quantity and quality needed to meet the consumer’s and market’s needs. Therefore, it is important to educate and train farmers about the methods of sustainable agriculture and to invest in advanced conservation agriculture techniques in developing countries [26].

Results of a study that was conducted in the Czech Republic and measured yielding parameters showed that ancient wheat species produced lower yields than modern wheat species. The experiment was conducted during two seasons at three locations with different soil and climate conditions. The experiment was composed of twelve plots, where *T. monococcum* (var. 65 Rumona), *T. dicoccum* (var. Rudico), and *T. spelta* (var. Rubiota) were sown in a completely randomized block design. No fertilizers were used during the experiment with ancient wheat species. To evaluate differences between old wheat species and modern planted wheat, samples of *T. aestivum* (var. Etana) from conventional growing were collected. All the ancient wheat species produced comparable yields and yielding parameters, and these were mainly affected by the soil and climate conditions of the experimental areas. Ancient wheats are well-known for their ability to grow naturally and simply without pretensions and their good level of resistance to diseases, which makes them favorable, sustainable, and ecofriendly. They can be grown under unfavorable soil and climate circumstances where they are more resilient to biotic and abiotic challenges than modern wheat, including disease, pests, drought, heat, cold, salt, pollution, and nutrient deficiency in the soil [27]. For example, in spite of the fact that the yield obtained from each cultivar of spelt wheat was lower than for common wheat, hulled wheat yields remained constant despite fluctuating agro-meteorological circumstances [28]. According to a German study, einkorn, emmer, and spelt, as ancient wheat varieties, had lower grain yields due to their tall plant heights, which made them prone to lodging. Their protein quality was also different from high-quality bread wheat. However, the way traits were inherited, with genetics and environment playing a role, was similar across all the ancient wheat species. Interestingly, the relationships between different traits in einkorn, emmer, and spelt were comparable to those seen in bread wheat. This suggests that improvements in the agronomic performance of einkorn, emmer, and spelt can be expected through plant breeding in the future, focusing on addressing their specific challenges. In terms of grain production, the mean emmer and einkorn yield was significantly lower than that of common wheat [23]. Raising awareness and interest on a global scale is needed to utilize organic and sustainable farming techniques and maximize grain production in low-input cultivation by cultivating cereals that are not influenced by environmental conditions such as drought, decreasing rainfall, fungus, and soil fertility [29]. Another study was conducted in the UK, and the results indicate that environmental factors affect the yields of spelt and emmer. In general, spelt had higher yields compared to emmer, except in certain parts of southern Britain and particularly in warm years. However, these interactions between yield and environmental factors did not fully explain why people chose to grow different wheat crops across Britain. The study only proposed that the shift from growing emmer to spelt might have occurred due to changes in cultivation practices during that time [30].

## 9. Countries Involved in Ancient Wheat Cultivation

Wild wheats were identified multiple times in the northern regions of Mesopotamia and uncultivated areas northwest of Anah, situated on the right bank of the Euphrates River. In 1877, wild spelt wheat was observed growing naturally north of Ramadan in western Iran. These findings led researchers to infer that the cultivation of wheat originated in the Euphrates Valley, which is primarily located within the central portion of the wheat cultivation belt spanning from China to the Canary Islands. Furthermore, the Euphrates basin, and possibly Syria, was the distribution area of wild wheat varieties in prehistoric times [18].

The majority of the available evidence suggests that ancient wheat farming was first practiced by humans in the Northeast, in the hills above the Tigris River on the western border of what is now Iran, where this territory has so many ancient agricultural sites. Emmer and einkorn wheat were among the first crops cultivated in this area. Then, the grain culture spread rapidly to Europe, West Africa, and the Nile Valley and grew all across the Caspian belt [31].

Emmer wheat has been largely cultivated during the 7th millennium in the Middle East, western and central Asia, and Europe, but is mostly found in Israel, Jordan, Lebanon, Syria, and Turkey, and is still found in this area. During the Neolithic and Bronze ages, emmer wheat was the primary wheat in the Old World and played a crucial role in the human diet of ancient civilizations, including the Assyrians, Babylonians, and Egyptians [32]. In most European countries, it was cultivated until the middle of the 20th century, primarily in mountainous areas; however, it is now a minor crop after being mainly replaced by durum and common wheat species, except in some countries where they still use emmer wheat grains to make traditional foods, such as India, Ethiopia, and Yemen, where it is still cultivated and highly valued today [33,34].

Einkorn was first cultivated around 9900 to 10,600 years ago in Southeast Anatolia, in Karaca Mountain. Then, it spread to the Caucasus, the Balkans, and Central Europe from the northern portion of the “Fertile Crescent” region. The Balkan nations—Spain, Germany, Italy, Switzerland, France, and Morocco—are still places where it is planted today on marginal agricultural territory, in addition to North Anatolia [35]. Currently, it is widely grown in northwest Turkey, and einkorn has popularity in the local Turkish market [36].

Spelt is still a significant cereal crop in isolated areas of southeastern Europe, mainly in Germany and Switzerland. According to some field experiments conducted in Italy, spelt yields were lower than those of bread or durum wheat but were intermediate between einkorn and emmer. While einkorn was historically grown in Italy’s harsh climate and poor soil, emmer is a minor wheat in Italy [37].

Khorasan, also known as oriental wheat, was primarily cultivated in specific regions such as Turkey, Iraq, Iran, Kazakhstan, Afghanistan, and parts of North Africa, including Egypt and Morocco. Notably, a collection of kamut landrace genetic resources was discovered in the Kahramanmaras region of Turkey [38].

In a separate study conducted in the Marchfeld region, northeast of Vienna, khorasan wheat and modern durum wheat were examined over a period of four years. The crops were sown both in the autumn and the spring. The findings indicated that the investigated khorasan wheat varieties generally displayed inferior agronomic traits compared to modern durum wheat. They exhibited limited tolerance to soil temperatures below −5 °C, drought, and fungal diseases. Additionally, their grain yields were significantly lower than those of durum wheat. Despite these limitations, khorasan wheat showed some interesting features that make it suitable for marketing, such as its large kernel size and high thousand kernel weight (often exceeding 50–60 g). These traits could be valuable for improving grain yield in durum wheat. Furthermore, khorasan wheat grains have an amber color and a glass-like appearance, which adds to their appeal. Despite the agronomic limitations observed, the unique qualities of khorasan wheat made it a promising option for marketing, either as a pure grain or as a component in grain blends [39].

In the work of Arzani and Ashraf (2017), they provided a map showing ancient wheat cultivation in ancient agricultural communities [5].

It is worth noting that the cultivation of other ancient wheat species, such as Caucasian (*Triticum timopheevii* and *Triticum zhukovskyi*), continues to be practiced to a limited extent in the Caucasus region for the purpose of producing traditional foods [40].

## 10. Nutritional and Health Benefits of Ancient Wheats

Ancient wheat varieties have multiple health benefits, including lowering glucose, insulin, lipids, and inflammatory risk factors. Chronic diseases, including cardiovascular diseases, diabetes, and obesity, are causing poor life quality and increasing mortality rates worldwide, and this is mainly related to poor nutrition. Therefore, researchers looked at the staple food for the majority of humans, which is wheat, and the findings of clinical implications from human studies proved the anti-inflammatory and antioxidant capacity of ancient wheat species. An example of that is the organic khorasan wheat products that have a positive impact on blood insulin, glucose, low-density lipoprotein (LDL) cholesterol, and the prevention of vascular diseases [19,41,42].

People who experience digestive issues after consuming common wheat products typically do not experience those symptoms after consuming hulled wheat products, and people who are allergic or intolerant to modern wheat are reported to tolerate some hulled wheat varieties better. Moreover, hulled wheat grains such as spelt, emmer, and einkorn grains are covered and protected by glumes, which are the main source of essential micronutrients and macronutrients [28]. Ancient wheat proteins are not suitable for manufacturing leavened baked products, but they do provide a different option for those who need to reduce their consumption of gluten overall. In addition, ancient wheat grains have a rich chemical composition and produce good-quality baked goods [27].

In another study, different health potentials were found in emmer, einkorn, and spelt wheats that proved to have antioxidant capacity combined with significant levels of dietary fibers, including arabinoxylans (AX). Higher total phenolic acid content, including ferulic acid, is a driving factor in the ultimate benefits of these nutrient-rich wheat species [43].

### 10.1. Analytical Methods in the Investigation of the Nutritional Properties of Ancient Wheats

The present subchapter provides a concise overview (Table 1) of the main analytical methods applied in the research on the nutritional properties of ancient wheats. Various techniques were reported to have been applied in the analysis of ancient wheat samples for the determination of their elemental composition, protein content, fiber, starch, fat, moisture content, or other phytochemicals, such as carotenoids or phenolic compounds.

Several classical methods, such as the Association of Official Analytical Chemists (AOAC) method based on the Kjeldahl method, micro-Kjeldahl nitrogen analysis (International Association for Cereal Chemistry, ICC method), and the American Association of Cereal Chemists (AACC) method specific to cereal grains, were used to measure protein content in different wheat samples. In addition, other techniques, including the Dumas method and near-infrared (NIR)-based measurements, were also applied. Different conversion factors were used to determine the protein content, with N × 5.7 being the most commonly used factor. However, other conversion factors, such as N × 5.75 and N × 6.25, have also been utilized in other studies.

The selection of the appropriate conversion factor is crucial for accurate protein determination, as it directly affects the calculated protein content. Researchers need to carefully consider the specific characteristics of their wheat samples and consult the relevant literature to determine the most suitable conversion factor for their analysis.

The Soxhlet method was widely used as the standard technique for fat content analysis, as described in ICC 1995 Method 136. Acid hydrolysis and subsequent Soxhlet extraction were generally applied.

For dietary fiber analysis, the enzymatic-gravimetric method using a commercially available kit was widely used to specifically target different types of fibers, including total, soluble, and insoluble ones. The ICC 1995 Method 136 and fiber analyzer were used for crude fiber determination.

Several methods were applied to measure the starch content of ancient wheat varieties, including enzymatic hydrolysis, the Ewers method, and the dinitrosalicylic acid (DNS) spectrophotometric method.

The moisture content in ancient wheat varieties was commonly measured gravimetrically by using the oven drying method, which involves drying a sample at 130 °C for two hours, followed by further drying at 105 °C until a constant weight is achieved. In addition to this, the thermal balance method was also utilized for moisture content determination by heating the sample at 120 °C.

Many methods were employed to analyze the ash content of ancient wheat samples, these being different in terms of temperature and duration of incineration, including AACC method 08-03, AACC method 08-01.01, and ICC.

For the analysis of minerals, atomic absorption spectroscopy, flame emission spectrophotometry, and inductively coupled plasma (ICP) atomic emission spectrometry were used, while X-ray fluorescence was utilized for the measurement of zinc and iron. Colorimetry was used to measure phosphorus, and inductively coupled plasma mass spectrometry was used for selenium.

High-performance liquid chromatography (HPLC) was employed for the determination of total tocopherols, tocopherols, and tocotrienols, while spectrophotometry was used for determining total carotenoid content.

For the analysis of other amino acids, different methods were applied, including an automatic amino acid analyzer, the Beckman System Gold method, phenylisothiocyanate (PITC) derivatization, and liquid chromatography coupled with the mass spectrometry (LC-MS/MS) method.

### 10.2. Nutritional Properties of Ancient Wheats: Macronutrients and Related Properties

This subchapter delivers information on the most representative macronutrients, including fat, protein, carbohydrate, starch, and dietary fiber. As nutritionally related parameters, moisture and dry matter contents, as well as ether extract and energy content, are also shown here (Table 2). The data is reported for four different types of ancient wheat (spelt, emmer, einkorn, and kamut) and common wheat. Table 2 contains a combination of numbers and descriptive terms to illustrate the changes in the macronutrient contents according to the type of wheat species. In the table, different colors denote data belonging to the different references marked with the same color.

Overall, by far the most common studied macronutrient in all wheat species was the protein content, and it was applied to different types of samples such as grain, flour, whole flour, endosperm, bran, germ, pasta, and cooked pasta. The single highest percentage of protein content was 48.5 ± 2.26% in the germ of einkorn wheat, while the lowest proportion was 11.0 ± 0.26% in common wheat grain. A further feature that can be mentioned is the difference in the applied conversion factor in the studies that used the Kjeldahl method to measure the protein content, which were 5.7, 5.75, and 6.25.

The percentage of each macroelement is reasonably similar across the five wheat species. On the other hand, it is interesting to note that total carbohydrate is the major component of wheat, and starch is the primary storage of carbohydrates, composed of about 60–75% grain and 70–80% flour [58]. In terms of resistant starch (RS), RS type 1 is found in grains, bread, and pasta, but accurate data on resistant starch intake is still not available [59]. RS has potential impacts on the prevention or therapy plans for metabolic diseases such as diabetes and obesity. It also reaches the large intestine and acts as a substrate for microbial fermentation, creating more good bacteria [60]. Data on resistant starch was only found in cooked pasta made from einkorn wheat (0.276–0.8%); therefore, further research needs to be conducted on the other ancient wheat species.

Regarding the fiber content, it is worth noting that einkorn and emmer grains contain a higher crude fiber content, reaching 5.19 and 5.03%, respectively, in comparison with the other wheat species.

**Table 2 foods-12-02411-t002:** Content of macronutrients and related parameters in ancient and common wheats.

Macronutrients	Spelt	Einkorn	Emmer	Kamut	Common Wheat	References
Moisture/Water (% or g/100 g)	26.8 ± 1.3 (bread) 9.1 (grain) 12.5 ± 4 (flour) 10.8 ± 0.14 (pasta) 15.04 ± 0.88 (flour)	13.7 ± 0.1 (flour) 9.84 ± 0.02 (pasta)		28.0 ± 1.4 (bread) 12.6 ± 4 (flour)	14.2 ± 4 (flour) 15.94 ± 0.36 (flour)	[61] [46] [47] [52] [54] [53]
Dry matter (% or g/100 g)	90.7 ± 0.85 (grain) 87.36 87.77 87.44 87.76 (grain) 88.40 88.58 88.80 89.16 88.20 88.65 88.76 88.18 88.41 (wholemeal)	90.4 ± 0.85 (grain)	90.9 ± 0.85 (grain)		90.5 ± 0.85 (grain)	[28] [62] [48]
Ether extract (% or g/100 g)	2.17 ± 0.03 (grain) 1.85 1.95 2 2.01 (grain)	2.43 ± 0.03 (grain)	2.28 ± 0.03 (grain)		1.72 ± 0.03 (grain)	[28] [62]
Energy (Kcal/100 g)	280 (bread) 324 (grain)			277 (bread)		[61] [46]
Fat (% or g/100 g)	1.8 ± 0.2 (bread) 2.5 (grain) 1.98 ± 0.2 (flour) 2.57 2.81 3.07 3.03 2.78 3.03 3.01 2.91 3.08 (wholemeal) 1.87 ± 0.021 (pasta)	1.87 ± 0.02 (pasta)		1.7 ± 0.2 (bread) 1.53 ± 0.3 (flour)	1.33 ± 0.1 (flour)	[61] [46] [47] [48] [52] [53]
Protein (% or g/100 g)	12.8 ± 0.31 crude (N × 5.7) > (grain) 15.9 ± 0.3 16.2 ± 0.3 17.1 ± 0.4 (N × 5.75) (seed) 12.7 < (N × 5.7) > (grain) 14.8 ± 1 (flour) 11.22 11.01 11.08 12.42 < crude protein (N × 6.25) > (grain) 13.9 ± 0.05 (pasta) 15.17 ± 1.13 (flour)	18.1 ± 0.41 crude (N × 5.7) > (grain) 15.8 ± 0.05 24.2 ± 0.45 11.1 ± 0.02 (whole flour) 13.8 ± 0.02 22.3 ± 0.11 10.1 ± 0.07 (endosperm) 8.2 ± 0.18 26.2 ± 0.54 14.2 ± 0.10 (bran) 48.5 ± 2.26 45.4 ± 2.75 22.4 ± 2.32 (germ) 13.1 ± 0.0 11.2 ± 0.0 16.9 ± 0.1 22.3 ± 0.1 (kernel) 20.7 ± 0.1 (flour) 20.2 ± 0.3 (pasta) 19.10 ± 0.07 18.3 ± 0.2 13.3 ± 0.2 (cooked pasta)	15.4 ± 0.39 crude (N × 5.7) > (grain)	17.8 ± 1 (flour)	11.0 ± 0.26 crude (N × 5.7) > (grain) 13.8 ± 0.3 (seed) 12.6 ± 1 (flour) 11.58 ± 0.28 (flour)	[28] [45] [46] [47] [62] [52] [54] [50] [51] [53] [56]
Carbohydrate (% or g/100 g)	71.9 ± 2.90 TC (grain) 49.7 ± 1.7 (bread) 62.7 TC (grain) 69 TC (flour) 70.77 ± 1.24 TC (flour)	62.3 ± 2.39 TC (grain)	65.9 ± 2.46 TC (grain)	49.9 ± 1.7 (bread) 67 (flour)	74.5 ± 2.99 TC (grain) 71 (flour) 67.78 ± 1.33 (flour)	[28] [61] [46] [47] [54]
Starch (% or g/100 g)	69.9 ± 0.89 (pasta) 64.21 ± 0.38 (flour)	53.3 ± 0.1 48.4 ± 0.1 55.9 ± 0.3 46.2 ± 0 (kernel) RS, resistant starch 0.276 ± 0.002 0.80 ± 0.02 0.382 ± 0.005 (cooked pasta)			66.38 ± 1.16 (flour)	[52] [54] [51] [56]
Dietary fiber (% or g/100 g)	1.96 ± 0.08 crude (grain) 6.2 ± 0.6 fiber (bread) Total: 13.8, Soluble: 1.7, Insoluble: 12.1 Total: 13.0, Soluble: 1.8, Insoluble: 11.2 Total: 12.9, Soluble: 1.7, Insoluble: 11.2 (seed) Total: 11.2, Soluble: 1.6, Insoluble: 9.6 (grain) 3.6 3.38 3.06 3.28 crude (grain)	5.19 ± 0.11 crude (grain) 10.1 ± 0.3 10.03 ± 0.08 3.6 ± 0.3 fiber (cooked pasta)	5.03 ± 0.10 crude (grain)	5.9 ± 0.6 fiber (bread)	1.78 ± 0.08 crude (grain) Total 12.8 Soluble 1.4 Insoluble 11.4 (seed)	[28] [61] [45] [46] [62] [56]

### 10.3. Nutritional Properties of Ancient Wheats: Micronutrients and Phytochemicals

The mineral content, reflected in ash, and the typical amounts of 20 elements reported in four ancient wheat species (spelt, emmer, einkorn, and kamut) and in common wheat were surveyed (Table 3). Overall, common wheat flour contains a lower total mineral content (0.063%) compared to ancient wheat flours (0.69–1.95%), while einkorn bran showed the highest ash content (7.5%). These values reveal the potential of ancient cereals in terms of their generally higher mineral content.

Macroelement content (Na, K, P, Mg, and S) showed uniform distribution in all the species analyses; however, iron contents were different depending on cultivar and sample type, with values ranging from 26.4 mg/kg in whole meal flour of spelt wheat to 52.35 mg/kg in spelt grains. Likewise, zinc content ranged from 22.9 mg/kg to 56.73 mg/kg in spelt grains, while common wheat grain had the lowest zinc content of 21.4 mg/kg. It was reported that both Fe and Zn correlated with P and S concentrations and with the protein content of the grains. The extent and strength of these correlations may vary depending on the specific wheat lines and conditions studied. The correlation between Fe and P suggests that higher levels of phosphorus in the grains are associated with higher concentrations of iron. Similarly, the correlation between Zn and P indicates that zinc concentrations tend to be higher in grains with increased phosphorus levels. Additionally, the correlation between Fe and S and between Zn and S suggests that higher sulfur concentrations in grains are associated with higher iron and zinc levels. As regards selenium, it was shown that genotypic differences are likely to be small compared to the variation in Se availability in the different soils. Thus, biofortification of wheat species with Se can be effectively achieved by applying Se-containing fertilizers. Einkorn, emmer, and spelt wheats showed higher concentrations of Se in grain than bread and durum wheats [3].

The germ fraction of spelt wheat showed the highest concentration of lutein, as high as 38 mg/kg; however, spelt lutein content ranged widely among cultivars. The lowest lutein proportion was found in common wheat (0.19 mg/kg). Similarly, the total carotenoid content of common wheat grains remained at just under 3.21 mg/kg, while in spelt flour it reached 8.13 mg/kg, and the germ fraction showed the highest concentration of protein and lutein in spelt wheat.

### 10.4. Nutritional Properties of Ancient Wheats: Amino Acid Composition

The literature data available on ancient wheats (Table 4) has proven that ancient wheats are scarce in some essential amino acids, particularly lysine and threonine. However, two essential amino acids (proline and glutamine) are abundant in wheat and responsible for dough-making. The proteins from spelt differed somewhat in their amino acid composition from those of common bread and pasta wheats [63]. According to our findings, it is worth noting that the general trend in the data was rather consistent regarding the majority of essential and non-essential amino acids in spelt wheat cultivars, except in the data reported by Ranhotra et al. [46], where slightly lower values were found for glutamic acid, valine, isoleucine, leucine, tyrosine, and arginine, but higher histidine values were found. The essential amino acid content was higher in einkorn lines than in emmer wheat lines and control cultivars. Therefore, the einkorn grain can be selected as a rich raw material for human nutrition due to the quality of its protein and amino acid contents.

## 11. The Aim of Sensory Tests in Cereal Research

There are two major objectives of any sensory test, which are considered almost central dogmas. The first is the analytical approach, when one wants to test either differences or intensities of attributes across the samples [64]. The second is the hedonic or preference approach, when the level of liking is measured, usually on a scale or with scores [65]. This latter test should only be performed with consumers, but in several papers, we find expert panels using five-point scales, where 5 is the best and 1 is the poorest. This approach probably comes from industry or trade quality control tests, but in research, it is strongly recommended to involve consumers in liking tests [66].

### Special Sample Preparation Issues for Sensory Tests

In many food research projects, sensory analysis is only part of the series of measurements made. Therefore, it is important to consider the special considerations that ensure the validity of these tests and their results from a sensory point of view. One of the basic parameters is the necessary number of samples for test sessions. Quite often, for analytical measurements, only minor quantities are sufficient. In the case of sensory tests, the necessary amount makes it possible to evaluate all the relevant attributes of the samples. In the case of bread, it is preferable to provide a whole slice [67], or if it is not possible, then a half slice per person. For example, in a French study that was conducted with einkorn, emmer, and spelt, the bread was cut into slices that were 13 mm thick [68].

Another issue is the preparation of cereal-based products and their storage before the session. In cases where sample preparation and testing are not on the same day, we have to consider how to store the samples to preserve their original attributes, especially the texture characteristics.

For example, the preparation of spelt wheat bread samples for sensory analysis was achieved in two steps:

Step 1The bread loaves were partially baked for 22 min, cooled for 2 h at room temperature, and then frozen at −30 °C. When it was fully cooled, Callejo and co-workers (2015) stored the bread samples in sealable plastic bags between the baking and serving days in a freezer at −18 °C with an air speed of 1.5 m/s. This technique minimizes the loss of volatile compounds and the drying of the texture.Step 2Before sensory evaluation, the samples were taken out of the freezer, left at room temperature for 1 h, and then fully baked for about 16 min in the oven at 210 °C. Finally, they were cooled down for 2 h at room temperature and then sliced properly for the sensory analysis [67].

Another study conducted a sensory evaluation test on 12 samples of commercial spelt bread, and sample preparation included warming up the samples. The 12 samples were divided into two groups of six, with the first group being served before a 10-min break and the second group being served after the break [69].

When the sample is sliced or served to the panelists, homogenous portions should be provided. This can be achieved by machinery (e.g., a bread cutting machine) or when the same person makes the serving [70]. Palate cleansing is used to help discriminate among samples, and the most frequent types are crackers and water [71], but in some papers, there are also other approaches using peeled cucumber, as in Starr’s work, to remove the carry-over effect [72].

## 12. Trained Panel and Expert Test Methods

In analytical sensory studies, the major focus is to characterize the samples with a comprehensive list of attributes and measure their intensities. The most frequent methods are based on sensory profile analysis (ISO 13299:2016) [73]. Some of these protocols are copyrighted, such as Quantitative Descriptive Analysis (QDA) or the Spectrum Method. In the case of any of those methods, the critical parameters are the theoretical and practical definitions of the attributes and the quality of the applied panel. Depending on the complexity of the study, the list of sensory attributes might be selected from previous studies, generated by a small number of experts, or collected by the panel during dedicated sessions [74]. Callejo and co-workers (2015) gave a very detailed example of the selection and training of panelists. In their work, they included basic selection tests (taste recognition and taste threshold tests) and a product-specific attribute evaluation (ranking of bread crust colors). This was followed by a series of paired comparison tests in order to differentiate between similar bread types. The final step of the training was a validation step, when triangle tests were used to confirm the discrimination capability of the individual panelists and also the whole panel. During the selection phase, nine panelists were selected from the initial 18 candidates. The four-month training period involved weekly sessions for the assessors. This approach is feasible in those cases when a detailed research project covers the expenses and the availability of panelists and panel leaders [67].

In an industrial environment, formalized training is replaced by a higher level of expertise from practical product evaluation. The major outcome of this study was the establishment of sensory profiles related to wheat and spelt breads. Crumb cell homogeneity and crumb elasticity showed the largest differences between the species. An important conclusion of the paper is that sensory attributes should also be considered when genetic breeding programs are designed [75].

Lomolino and co-workers (2017) investigated the bread quality prepared from einkorn and common wheat. In their study, several investigations were performed, including very special imaging techniques (X-ray microtomography), so the sensory tests were not the dominant part of the project. This is a very good example of such research projects when the authors aim to give a complex analysis of the samples. In this paper, the chosen method (QDA) is a suitable choice to understand the differences between raw materials and fermentation treatments. Most of the attributes conform to the profiling method since they can be measured on an intensity scale (color, flavor, texture, and firmness). Simplified quantitative descriptive analysis (QDA) is mainly used because there are only a few attributes and they are quite general in nature. This approach uses scoring tests, where panelists have to give scores to four or five attributes, and the summarized scores indicate the quality category of the product. In this paper, the last sensory attribute confirms this approach since it was called “overall quality”, and later in the text “overall acceptability”. This approach is rather typical in those studies, when researchers aim to compare the newly developed samples to typical bread styles (true to style), or they look at the samples as a quality control panel, searching for possible technological faults [76]. A similar approach was used in another study, where the analyzed attributes are rather specific (appearance of crumb grains, mouthfeel of crumb, flavor intensity of crumb, and firmness of crust), but the applied method relies on a 9-point hedonic scale. The panelists are trained food scientists. In this latter case, the selection of attributes is more specific to the research field, which shows the technological expertise of the project leaders. However, they did not adopt the analytical thinking of profiling or QDA tests in their study [77].

Starr and co-workers (2013) implemented a special study, preparing cooked ancient wheat grains in accordance with the New Nordic Diet. The total number of tested species and varieties was relatively high (24 samples altogether). However, the researchers applied an effective strategy of balanced incomplete blocks. Through that layout, seven to nine samples were served to the panelists at a single session, plus a reference sample. The selection and training procedures were based on the relevant ISO standards. The panelists created a common list of attributes during the training sessions. For most of the descriptors, suitable reference material was provided [72].

## 13. Affective Tests, Size of Consumer Panels

Bagdi and co-workers (2016) analyzed the quality of bread samples made from aleurone-rich flour. Their study involved a descriptive sensory test (profile analysis) with a trained panel, followed by a consumer test with 80 participants. In academic research studies and pilot projects, there is a guideline in ISO 11136 that recommends a minimum of 60 participants. It is also specified in that standard that the smallest segment of consumers should not be smaller than 60 people. In cases where the experimenter plans to form clusters (based on socio-demographic parameters or personal preference), the number of involved panelists rises dramatically. In this paper, the researchers applied two types of scales for data collection. The first was a 9-point hedonic scale, which is generally accepted in consumer studies. This measures the degree of liking and shows the relative differences between the samples. In the second step, the consumers worked with optimum scales (just-about-right, JAR scales). This approach defines the middle of the scale as the optimal intensity of a given attribute. If the consumers give a lower or higher score than the middle point, it means that the attribute is too strong or too weak. This dataset is analyzed with penalty analysis, which identifies those product attributes that highly influence the preferences of consumers [78].

Hersleth and co-workers (2005) implemented a consumer test and descriptive profiling in parallel. Their research goal was to investigate the perceptions of bread quality among consumers and bread experts. It is often debated, even in food science papers, whether consumer preferences or expert opinions should be followed when formulating a new product type. In their study, they involved 30 consumers. The major criteria was that they were regular users of bread. The experimenters carefully balanced the gender ratio (50% each). The method they applied is called the repertory grid, which is a special approach that originated from psychological experiments. Participants were given sample triads, and they were asked to choose the sample that was most different from the other two, and they had to give an argument about the nature of the difference. With that technique, each consumer created their own construct, or, in other words, a list of attributes that helped them differentiate between the seven bread samples. In the final part of the study, the experimenters edited an individual test sheet for each participant and asked them to rate the bread sample intensities on a 9-point scale. Notably, this scale was not based on preferences but on intensities. The trained panel worked with the QDA method. The 11-member panel created a remarkably high number of attributes (*n* = 38) and analyzed the sample intensities. This test was performed in a standardized laboratory environment with two replicates for each sample [79].

In another **study** (Škrobot et al., 2022), **emmer,** spelt, and khorasan flours were used to prepare sourdough bread. In addition to performing a sensory profile analysis (involving 29 descriptors), the authors also implemented a consumer test. The size of that panel exceeded the recommendation in ISO 11136 (a minimum of 60 people) since it contained 72 participants. The test sheet contained four preference questions on overall opinion, taste, texture of the crumb, and texture of the crust. The level of preference was measured on a 9-point hedonic scale. Panelists were recruited from university students and staff, which is a frequent technique in academic research. At that level of test participants, the total data might be segmented with hierarchical cluster analysis (HCA), but that was not involved in that study [80].

Korciari and co-workers (2021) investigated the effects of different fermentation strategies on the quality of spelt breads. They prepared four bread samples with different treatments. In the original research plan, a wheat-based control sample was also involved, but in the consumer testing part, this sample was omitted since it showed a large degree of difference from the other specimens. In the study, 86 participants were recruited from the students and staff of the institute. The authors have referred to the number of consumers, stating that their number is not as high as in representative market research projects but large enough to meet the requirements of pilot testing described in ISO 11136. Participants analyzed a slice of each bread sample, placing them on a 100-mm unstructured line scale of overall acceptance. Water was provided as a palate cleanser. Samples were coded with 3-digit random codes, and their presentation order was randomized and balanced [81].

## 14. Sensory Room Environment

The accuracy and reliability of sensory evaluations are strongly influenced by the sensory room environment, which plays a crucial role in ensuring the consistency and reproducibility of sensory data. Sensory rooms are designed to provide a controlled and standardized environment that minimizes potential sources of bias and variability, such as noise, lighting, temperature, and humidity [82].

In a Danish study aimed at providing valuable information about the sensory profiles of cooked grains, specifically related to bread, the sensory evaluations were conducted in a laboratory specifically designed for this purpose according to the ISO 8589:2007 guidelines. Each assessor was assigned to their own booth, which was equipped with air extraction, an independent light source, and maintained an ambient temperature of 22 °C. Sensory evaluation involved trained sensory panels assessing and rating the cooked grains to understand the differences and similarities among the wheat species and varieties in terms of their potential for breadmaking [72]. Similarly, in another study carried out in Serbia, the sensory assessment was conducted in a laboratory designed for testing purposes, and the environmental conditions were controlled according to ISO 8589 General Guidance for the Design of Test Rooms [83]. In a French study, however, the sensory profile training and evaluation were carried out by the panel members in a meeting room located at the INRA laboratory, Le Moulon, which is a French National Institute for Agricultural Research [68].

Nevertheless, according to a study in the USA, the sensory evaluations were conducted in individual booths that were well lit, ventilated, and isolated from external odors and noise, without mentioning the exact sensory room guidelines used in this study [84].

## 15. Statistical Analysis of Sensory Data

Most of the studies started with the implementation of a univariate statistical test, usually an ANOVA (analysis of variance). In some cases, the exact equations of the applied protocol are also provided [72]. In some research papers [81], the normality of the data was also investigated prior to ANOVA, since from a statistical point of view, this is a prerequisite of that protocol. Data normality can be investigated with several statistical methods, such as the Shapiro–Wilk test. Significant differences among the samples are labeled in the tables and diagrams with different letters following the average values. This labeling relied on post-hoc tests, of which the most typical ones are the Bonferroni *t*-test, Tukey-test, and Fisher’s Least Significant Differences. In those studies where QDA or similar descriptive tests were applied, the complex data matrices provided a good starting point to perform a multivariate test. The most typical is the Principal Component Analysis (PCA), but we also find correlation protocols (Pearson’s correlation coefficient) and more specialized approaches, such as the Generalized Procrustes Analysis [79]. PCA provides an effective synergy to analyze sensory and instrumental data together and identify possible internal relationships in the data matrices. Concerning the applied statistical software choices, some of the most frequently applied are SPSS, SAS, Matlab, XLStat Sensory, and also scripts in the R programming language.

### Comparison of Sensory Evaluation Methods for Different Ancient Wheat Products

In terms of sensory attributes for ancient wheats, spelt grain has a rich flavor described as sweet and nutty; the pleasant texture of baked products and rich nutritional profile are said to be the reasons why consumers are interested in it. Numerous products can be made from spelt, primarily leavened bread [29], especially since spelt flour is characterized by moderate technological value [85]. The cooked, whole, or dehulled einkorn kernel was subjected to sensory profiling, which showed that the samples had a softer texture and were less mealy, adhesive, and fibrous compared to spelt and common wheat samples. In terms of taste attributes, einkorn samples had a more pronounced “hot oatmeal porridge” taste than the wheat sample and were slightly sweeter than other samples. Overall, the einkorn samples were highly regarded for their pleasant consistency and flavor [86]. Cooked emmer grains were described as having a nutty and prominent flavor, and there were no significant differences among emmer varieties [84]. Increasing the emmer flour amount in the samples resulted in an increase in the brittleness value, and the color of the samples made with emmer flour was darker and more reddish compared to those made with einkorn flour [87]. The taste of khorasan is sweet and grassy and can be described as a combination of raw sweet corn and freshly cut lawn with a milky undertone [88]. An Egyptian study revealed that kamut couscous was highly favored by the participants, who described its taste as tasty, buttery, crunchy, and delicious, so it has many taste advantages [89]. Kamut is used in a wide variety of food products, including cereals, breads, cookies, snacks, pancakes, bread mixes, bulgur, pasta, and baked goods [90]. The sensory and physical properties of bread made with a blend of ancient cereals, including kamut and spelt, were found to be similar to those made with common wheat flour, so making bread mix from ancient wheats also offers a range of nutritional and sensory potential benefits [47].

The following section presents information about the sensory evaluation methods applied to different types of wheat products (Table 5). The category of bread shows that bread was made from different wheats, including spelt, einkorn, durum, bread wheat, and hard red spring wheat. Five studies applied five different sensory evaluation methods to bread samples, including a nine-point hedonic scale, a ten-point scale, an unstructured scale (10 cm straight line), a questionnaire, and a quantitative descriptive profile analysis (QDA). The evaluation panels varied considerably in size, ranging from 4 to 25 panelists, with some of them being trained food scientists and others being testers familiar with sensory analysis of food but not specially trained in the evaluation.

In terms of pasta, Table 5 shows eight studies that conducted sensory evaluation on pasta samples that were made from different wheats, including spelt, einkorn, emmer, and durum. Different sensory evaluation methods were used in the studies, including a nine-point hedonic scale, a five-point hedonic scale, a combination of ranking tests/hedonic scales, scores, and descriptive sensory analysis. The evaluation panels varied in size, ranging from 3 to 15 participants, and some of them were trained food technicians or experts in sensory vocabulary and identification, with others being semi-trained or not trained at all.

Other evaluated products include cooked grains, porridge, snacks, and muffin products made from different wheat varieties such as spelt, einkorn, emmer, durum, kamut, and modern wheat. Various sensory evaluation methods were used, including hedonic scales, descriptive tests, descriptive sensory profile analysis, and Likert scales. Evaluation panels varied in size, ranging from 3 to 26 participants, and in training, with some being trained assessors and others being volunteers without specific training.

## 16. Conclusions

The present review critically evaluates the nutritional and sensory properties of ancient wheats, including spelt, emmer, einkorn, and kamut, as well as the methods used to analyze them. Although numerous studies on the nutritional and health-related qualities of ancient wheat species have been conducted, full comparisons between these species and current wheat will be challenging due to environmental considerations. There is currently scarce information on the usage of ancient wheat flours as partial replacements for modern wheat flour in the creation of breads, biscuits, and pasta.

Incorporating ancient wheat varieties into food products may offer additional nutritional benefits and interesting sensory properties for consumers. Therefore, it is important to support local farmers who grow these varieties as sustainable and wholesome food supplies. Based on the analysis of available research, it can be concluded that the utilization of appropriate analytical methods is crucial in mapping the nutritional properties of ancient wheats. Understanding the sensory characteristics of ancient wheat varieties provides valuable information regarding consumer acceptance, preferences, and overall product quality. Unique breads with distinctive flavors and textures can be created using these cereals as raw materials, contributing at the same time to more biodiverse and sustainable agricultural and food systems. By considering the nutritional and sensory aspects, manufacturers and producers can create ancient wheat products that not only offer functional nutritional benefits but also deliver a satisfying sensory experience to consumers.

## Figures and Tables

**Table 1 foods-12-02411-t001:** Comparison of methods applied for analyzing components in ancient wheat samples quantitatively.

Measured Parameters and Methods Applied						
Cereals Involved	Sample Type	Macronutrients	Micronutrients	Amino Acids	Results	Reference
Emmer Einkorn Spelt Bread wheat	Whole grain	Five macroelements (**K, P, S, Mg, Ca**): ICP-SFMS measurements were performed on a double-focusing ICP-sector field	Eleven microelements and trace elements (**Zn, Fe, Mn, Na, Cu, Al, Ba, Sr, B, Rb, Mo**): ICP-SFMS measurements were performed on a double-focusing ICP-sector field		Potassium concentrations were significantly higher in the grains of spelt and emmer than in common wheat.	[44]
Einkorn Emmer Spelt Bread wheat	Grain	(**Total protein, crude fat, crude fiber, total ash, and total carbohydrates**): (AOAC method)	(**K, Ca, Mg, Fe, Mn, Zn, Cu**): atomic absorption spectrometer		The highest protein content was determined in einkorn wheat (18.1%), followed by emmer (15.4%), spelt (12.8%), and common wheat (11.0%).	[28]
Spelt Bread wheat Durum	Milled seeds	**Ash and moisture:**** (AACC, 1995)** methods **Protein:** (Kjeldahl method N × 5.7) (979.09) **(AOAC, 1990)** **Total dietary fiber (TDF):** enzymatic and gravimetric procedures **Starch:** enzymatic hydrolysis		Beckman System Gold	The spelt wheat cultivars studied had a higher protein content than the standard cultivars of common wheat and durum wheat. No difference was found in the ash content between spelt and common wheat.	[45]
Spelt	Grain	**Moisture:** Standard AACC (5) **Protein:** (Kjeldahl method N × 5.7) **Fat:** ether extract **Ash:** Standard AACC (5) **Fiber (total, soluble, and insoluble):** enzymatic-gravimetric method **Carbohydrate:** subtracting the sum of moisture, protein, fat, ash, and fiber content from 100	**Minerals (Ca, Cu, Fe, K, Mg, Zn, Na),** except phosphorus: atomic absorption or flame emission (sodium) spectrophotometry **Total phosphorus**: colorimetrically	**Amino acids, except tryptophan and cysteine:** phenylisothiocyanate (PITC) method	Spelt grain has a little higher total fat and digestible carbohydrates (starch and sugars), and it contains about 3% more energy.	[46]
Durum Spelt Einkorn Emmer Bread wheat	Grain		**K, Ca, Mg, S, P, Fe, Zn, Mn, Cu**: inductively coupled plasma (ICP) atomic emission spectrometry **Se**: inductively coupled plasma mass spectrometry		Spelt, einkorn, and emmer wheat grains contain higher Se concentrations than bread and durum wheats.	[47]
Spelt	Grain	**Crude protein**: (Kjeldahl method N × 6.25) **Ether extract**: Soxhlet extraction **Crude fiber**: fiber analyzer	**Crude ash**: incineration in a muffle furnace at 580 °C for 8 h	**Amino acids (methionine, tryptophane, leucine, phenylalanine, and total sulfur-containing amino acids):** AAA 400 automatic amino acid analyzer **Tryptophan:** method described in **AOAC (2006)**	There are positive correlations between Fe and Zn concentrations and protein content.	[3]
Spelt	Wholemeal Sieved flour Fine bran Coarse bran	**Total lipid**: Soxhlet method **Dry Matter**: desiccation of cereal samples, weighed and dried at 105 °C for 24 h, and then weighed again	**Ash:** 5 g samples; incineration at 550 °C for 16 h **All minerals (Ca, Mg, Fe, Zn, Cu, Mn)** except phosphorus: atomic absorption after the ashing **Na and K**: flame photometry **P**: colorimetry, AOAC method 995.11 **Total tocopherols**: HPLC		The nutritional advantages of spelt over wheat would be best expressed in wholemeal bread or bran nutrition bars rather than in bread from sieved or refined flours.	[48]
Spelt	Grain	**Protein**: Infratec 1241 Grain Analyzer	**Zn and Fe**: X-ray fluorescence		The grains of spelt wheat parents contained significantly more Zn and Fe than bread wheat.	[49]
Emmer Einkorn Durum	Grain	**Total protein**: Dumas method		Liquid chromatography tandem mass spectrometer (LC-MS/MS) method	Einkorn had the highest mean protein content, followed by emmer, whereas durum wheat had the lowest concentration of protein.	[9]
Einkorn	Whole flour endosperm Bran Germ	**(AACC, 1994)** **Dry matter:** 44–15 **Protein:** 46–10 (N × 5.7)	**Ash:**** (AACC, 1994)** 08–03 **Tocopherols and tocotrienols**: normal-phase HPLC		A nutritional advantage over the de-branned and de-germinated flour, mainly because of the high tocol content in the germ and bran.	[50]
Einkorn Bread wheat	Kernels	**Total starch**: Megazyme assay kit **Protein**: **(AACC, 1995)** 46–10 (N × 5.7)	**Carotenoid:** extraction and quantification by HPLC **Tocols:** HPLC		The einkorn cultivars showed a superior concentration of total carotenoids than the bread wheat in the raw kernels.	[51]
Spelt Einkorn Emmer Quinoa Durum	Raw (uncooked) pasta	**Moisture**: drying at 130 °C for 1.5 h in a forced air oven **Crude protein**: Kjeldahl method (N × 5.7) **Starch**: Ewers method **(ISO 10520, 1997)** **Fat**: acid hydrolysis and subsequent Soxhlet extraction	**Ash**: incineration in a muffle furnace for 4 h at 900 °C		The spelt wheat pasta had a higher ash content (1.10%) compared to the durum wheat pasta.	[52]
Einkorn	Flours Semolina Dry pasta	(AACC) **Moisture:** 44–15.02 **Ash**: 08–03.01 **Protein**: Kjeldahl method (N × 5.7) 925.31 **(AOAC, 1995)** **Lipid**: Soxhlet method 136 **(ICC, 1995)**	**Total carotenoid:** spectrophotometric method 14–60.01 (AACC)		Einkorn pasta had more protein than durum wheat.	[53]
Spelt	Flours	**Moisture**: (ICC methods 110/1) **Protein**: Kjeldahl method (N × 5.7) (979.09) **(AOAC, 1990)** **Starch**: dinitrosalicylic acid spectrophotometric method (DNS)	**Ash**: ICC standard No. 104/1		Spelt wheat flour had a higher protein content and exhibited different protein fractions and properties from common wheat.	[54]
Einkorn Emmer Rivet	Whole grains were stone milled (on a whole flour dry weight basis)	**Protein**: Foss Infratec 1229 NIT spectrophotometer **Lipid analysis**: AOAC method **Total dietary fiber**: enzymatic-gravimetric procedure using a Megazyme assay kit			The highest protein content was found in emmer, followed by einkorn. The highest fat content was detected for einkorn, followed by emmer, and appreciably higher than those reported in the literature for common wheat.	[55]
Einkorn	Whole grain Cooked pasta Freeze-dried pasta	**Moisture:** thermal balance at 120 °C. **Protein:** micro-Kjeldhal nitrogen analysis (ICC 105/2 method) (N × 5.7) **Resistant starch (RS):** Official Method 2002.02, using Resistant Starch Assay Kit **Total dietary fiber (TDF):** enzymatic-gravimetric kit for fiber determination according to the Official Method 991.43	**Ash:** AACC 08-01.01 method		Significantly higher protein contents were observed in einkorn pasta in comparison with those reported for durum wheat semolina pasta.	[56]
Spelt	Wholemeal flour Flour Pasta	Standard **(ICC 1995)** for: **Moisture:** (ICC Method 110/1) **Crude protein:** (N × 5.7) (Method 105/2) **Total fat:** (Method 136) **Soluble, insoluble, and total dietary fibers:** enzymatic gravimetric procedure of Prosky, Method 985.29 **(AOAC 1995)**	**Ash:**** (ICC 1995)** (Method 104/1)		Spelt samples (wholemeal flour) had fat contents significantly higher than those of common wheat.	[57]
Einkorn	Whole grain Cooked pasta Freeze-dried pasta	**Moisture:** thermobalance at 120 °C **Protein:** micro-Kjeldhal nitrogen analysis (ICC 105/2 method) (N × 5.7) **Resistant starch (RS)**: Official Method 2002.02, using Resistant Starch Assay Kit **Total dietary fiber (TDF):** enzymatic-gravimetric kit for fiber determination according to the Official Method 991.43	Ash content: AACC 08-01.01 method		Significantly higher protein contents were observed in einkorn pasta in comparison with those reported for durum wheat semolina pasta.	[56]
Spelt	Whole-meal flour Flour Pasta	Standard **(ICC 1995)** for: **Moisture:** (ICC Method 110/1) **Crude protein:** (N × 5.7) (Method 105/2) **Total fat:** (Method 136) **Soluble, insoluble, and total dietary fibers:** enzymatic gravimetric procedure of Prosky, Method 985.29 **(AOAC 1995)**	**Ash:**** (ICC 1995)** (Method 104/1)		Spelt samples (wholemeal flour) had a high fat content, significantly higher than that of common wheat.	[57]

**Table 3 foods-12-02411-t003:** Content of micronutrients in ancient and common wheats.

Macronutrients	Spelt	Einkorn	Emmer	Kamut	Common Wheat	References
Ash (% or g/100 g)	1.86 ± 0.07 crude (grain) 2.2 ± 0.3 (bread) 1.76 ± 0.02 1.82 ± 0.02 1.85 ± 0.01 (seed) 1.8 (grain) 0.161 ± 0.02 (flour) 1.97 1.98 1.86 1.98 (grain) 1.67 1.81 1.94 1.94 1.81 1.85 1.96 1.68 1.76 (wholemeal) 1.24 (pasta) 1.95 ± 0.41 (flour)	2.65 ± 0.08 crude (grain) 0.69 ± 0.01 (flour) 0.71 (pasta) 2.2 ± 0.01 2.5 ± 0.01 1.5 ± 0.00 (whole flour) 0.4 ± 0.01 0.7 ± 0.03 0.4 ± 0.00 (endosperm) 7.5 ± 0.03 6.8 ± 0.01 6.8 ± 0.01 (bran) 4.8 ± 0.11 4.4 ± 0.15 2.7 ± 0.11 (germ) 2.59 ± 0.01 2.26 ± 0.03 0.708 ± 0.001 (cooked pasta)	2.16 ± 0.07 crude (grain)	2.0 ± 0.3 (bread) 0.136 ± 0.05 (flour)	1.52 ± 0.06 crude (grain) 1.83 ± 0.01 (seed) 0.063 ± 0.01 (flour) 0.63 ± 0.09 (flour)	[28] [61] [45] [46] [47] [62] [48] [52] [54] [53] [50] [56]
Sodium Na (mg/kg)	10 (grain) 10 (grain) 6.10 6.46 9.22 8.93 10.92 7.46 7.07 8.30 11.89 (wholemeal)	7 (grain)	12 (grain)		10 (grain)	[44] [46] [48]
Potassium K (g/kg)	4.17 (grain) 4.11 (grain) 3.10 3.82 4.03 3.91 3.68 3.83 3.66 3.58 3.85 (wholemeal)	4.29 (grain)	4.39 (grain)		5 (grain)	[44] [46] [48]
Phosphorus P (g/kg)	4.7 (grain) 4.62 (grain) 3.173 2.819 2.445 2.969 2.813 2.845 2.959 3.317 2.965 (wholemeal)	5.2 (grain)	5.12 (grain)		4.18 (grain)	[44] [46] [48]
Magnesium Mg (g/kg)	1.5 (grain) 1.31 (grain) 1.2750 1.3331 1.3223 1.2986 1.3032 1.2560 1.2956 1.1881 1.1952 (wholemeal)	1.63 (grain)	1.67 (grain)		1.44 (grain)	[44] [46] [48]
Sulfur S (g/kg)	1.8 (grain)	1.93 (grain)	1.88 (grain)		1.4 (grain)	[44]
Iron Fe (mg/kg)	50 (grain) 38 (grain) 41.8 (grain) 34.7 30.7 28.4 36.5 28.5 28.8 38 27.9 26.4 (wholemeal) 51.4 51.99 51.37 49.58 42.73 52.35 51.65 46.86 44.68 51.9 (grain)	45.9 (grain) 49 (grain)	49 (grain) 34.1 (grain)		38.2 (grain) 37.5 (grain)	[44] [46] [3] [48] [49] [44]
Copper Cu (mg/kg)	5 (grain) 6 (grain) For all 9 cultivators: < 1 (wholemeal)	4 (grain)	4.1 (grain)		3.9 (grain)	[44] [46] [48]
Zinc Zn (mg/kg)	47 (grain) 42 (grain) 22.9 (grain) 30.9 29.7 29.7 31.7 31.9 29.8 35.1 25.8 29.8 (wholemeal) 56.73 54.64 54.32 54.2 53.9 53.87 52.69 51.64 51.29 51.14 (grain)	53 (grain) 22.4 (grain)	54 (grain) 22.8 (grain)		35 (grain) 21.4 (grain)	[44] [46] [3] [48]
Selenium Se (μg/kg)	790 ± 10 (bread) 125.1–244.0 (grain)	178.5–440.0 (grain)	150.6–325.8 (grain)	560 ± 20 (bread)	32.9–237.9 (grain)	[61] [3]
Manganese Mn (g/kg)	27 (grain) 31.9 28.5 26.8 27.8 29.1 29.2 28.2 26 28 (wholemeal)	28 (grain)	24 (grain)		26 (grain)	[44] [48]
Aluminium Al (mg/kg)	4.4 (grain)	2.5 (grain)	3.8 (grain)		1.7 (grain)	[44]
Rubidium Rb (mg/kg)	1.1 (grain)	0.8 (grain)	0.8 (grain)		1.45 (grain)	[44]
Strontium Sr (mg/kg)	3.6 (grain)	5.4 (grain)	2.6 (grain)		3.0 (grain)	[44]
Barium Ba (mg/kg)	3.5 (grain)	2.6 (grain)	2 (grain)		2.95 (grain)	[44]
Molybdenum Mo (mg/kg)	0.7 (grain)	1.2 (grain)	1 (grain)		0.65 (grain)	[44]
Boron B (mg/kg)	0.7 (grain)	0.8 (grain)	0.6 (grain)		0.75 (grain)	[44]
Total carotenoid (mg/kg)	8.13 ± 0.01 (flour) 4.79 ± 0.16 (pasta)				0.48 ± 0.01 0.89 ± 0.02 3.21 ± 0.04 2.01 ± 0.06 (kernel)	[53] [51]
Carotenoid (Lutein) (μg/g) or (mg/kg)	7.5 ± 0.02 5.8 ± 0.19 0.9 ± 0.09 (whole flour) 7.4 ± 0.15 5.1 ± 0.03 0.8 ± 0.01 (endosperm) 4.0 ± 0.08 4.5 ± 0. 0.7 ± 0.01 (bran) 38.0 ± 1.06 26.3 ± 3.83 6.3 ± 2.88 (germ)				0.19 ± 0.00 0.48 ± 0.01 1.04 ± 0.05 0.81 ± 0.04 (kernel)	[51] [50]

**Table 4 foods-12-02411-t004:** Content of amino acids in ancient wheat varieties.

Amino Acids	Spelt (g/100 g)							Emmer (g/100 g)	Einkorn (g/100 g)
Aspartic acid	5.2	5.3	5.65					24.22 ↔ 49.20	64.33 ↔ 150.38
Threonine	2.7	2.9	2.11	2.4	2.5	2.5	2.4	0.87 ↔ 5.32	1.81 ↔ 7.63
Serine	4.7	4.7	4.79					0.00 ↔ 33.26	0.00 ↔ 65.42
Glutamic acid	36.0	30.9	26.79					26.30 ↔ 66.10	50.74 ↔ 87.92
Proline	11.9	8.9	8.12					1.29 ↔ 30.84	4.30 ↔ 18.45
Glycine	3.8	4.4	4.92						
Alanine	3.4	3.6	3.45						
Cysteine	2.1	2.4						0.00 ↔ 2.00	0.00 ↔ 5.12
Valine	4.7	4.7	1.81	3.8	3.8	3.7	3.6	0.00 ↔ 31.84	0.00 ↔ 54.81
Leucine + Isoleucine								1.86 ↔ 5.75	7.80 ↔ 16.57
Methionine + Cysteine				3.1	3.1	3.1	2.9		
Methionine	1.7	2.0	1.33	1.6	1.6	1.7	1.5	0.39 ↔ 2.69	1.03 ↔ 4.06
Isoleucine	3.8	3.8	1.03	2.9	3.0	2.9	2.9		
Leucine	7.1	7.0	4.15	6.5	6.5	6.4	6.6		
Tyrosine	2.7	2.3	1.44	2.5	2.5	2.4	2.4	2.23 ↔ 7.71	6.34 ↔ 12.18
Phenylalanine	5.1	5.4	3.02	4.6	4.4	4.6	4.6	3.90 ↔ 6.89	6.67 ↔ 11.10
Lysine	2.7	2.8	2.04	2.6	2.7	2.6	2.4	14.97 ↔ 76.72	69.006 ↔ 231.30
Histidine	2.4	2.3	3.14	1.9	1.9	1.8	1.8	0.95 ↔ 2.90	1.79 ↔ 4.03
Arginine	4.5	4.5	2.38					0.05 ↔ 5.30	0.10 ↔ 4.45
Tryptophan				1.1	1.1	1.1	1.1		64.33 ↔ 150.38
References	[45]	[45]	[46]	[62]	[62]	[62]	[62]	[9]	[9]

**Table 5 foods-12-02411-t005:** Sensory evaluation methods of different types of ancient wheat products.

Cereals Involved	Product Type	Sensory Test Type	Number of Panelists/Consumers	Reference
Spelt Bread wheat	Bread	Questionnaire	25 participants	[91]
Spelt Bread wheat	Bread	**(ISO 6564, 1985; ISO 4121, 2003)**. The intensity of each attribute was scored on an unstructured 10-cm straight line labeled “not noticeable” and “very strong” at the left and right end points, respectively.	18 adults (nine females and nine males) aged between 27 and 65. Then, 9 members were selected and trained.	[67]
Spelt Einkorn Durum Hard red spring	Bread	Nine-point hedonic scale	12-member panel of food scientists trained in sensory evaluation	[77]
Spelt Common wheat	Bread	Scale scoring 0–10; evaluated by quantitative descriptive profile analysis (QDA)	4–9 trained panelists	[54]
Einkorn Common wheat	Bread	Ten-point scale (where “1” represented low intensity and “10” high intensity)	15 testers familiar with sensory analysis of food but not specially trained in the evaluation of sourdough breads; ages ranged from 22 to 40 years old (9 women and 6 men).	[76]
Spelt	Pasta (Spaghetti)	Five-point hedonic scale and then converted into numerical scores	Trained panel of 3 assessors	[92]
Einkorn	Pasta (Spaghetti)	Each sensorial parameter was scored from 10–100	Panel of 5 trained assessors, who are food technicians at the ‘Cereal Food Processing Lab’ in Rome	[56]
Einkorn Durum	Pasta	Each descriptor was scored from 10–100	Panel of 3 trained assessors	[93]
Spelt	Pasta	Ranking test/hedonic scale Nine-point hedonic rating scale	Panel of 7 experts, selected according to their sensorial skills and trained in sensory vocabulary and identification of particular attributes	[94]
Spelt	Pasta	Score of 0–100	Not available	[57]
Spelt	Pasta	Nine-point hedonic scale	15-member semi-trained panel (7 males and 8 females, ages 23–40)	[95]
Spelt Einkorn Emmer Common wheat	Pasta	Questionnaire	Group of 10 evaluators	[96]
Emmer	Pasta and cooked grains	Descriptive sensory analysis	12 trained panelists and 26 public preference tasters	[84]
Spelt Einkorn Emmer Kamut Modern wheat	Cooked grains	Descriptive tests/sensory profile analysis	10 experienced assessors (3 men and 7 women between the ages of 21 and 39)	[72]
Einkorn Durum	Uncooked and cooked bulgur samples	Ranking test/hedonic scale 5-point hedonistic scale (1: very defective, 3: acceptable, 5: perfect) Mixture of hedonic testing and quality scoring	10 well-briefed panelists	[97]
Spelt	Porridge made from whole grains	Questionnaire using a 5-point Likert scale from “extremely unpleasant” (1) to “extremely pleasant” (9), followed by a 5-point scale from “totally agree” to “totally disagree”.	Total of 129 volunteer Finnish women in 4 experimental groups	[98]
Spelt	Snack products made from whole grains	Unstructured linear scale from imperceptible (0) to very intense (100)	8 trained panelists, between 25 and 50 years old	[83]
Kamut	Muffin	Nine-point hedonic scale	51 panelists from Korean University (ages 20–60)	[99]

## Data Availability

The data presented in this study are available on request from the corresponding author.

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
