# Peer review of "Ancient Wheats—A Nutritional and Sensory Analysis Review"

_foods, 2023, doi:10.3390/foods12122411_

Round 1
Reviewer 1 Report
The present review article is focused on ancient wheats and sensory evaluation of breads and other wheat products. Nutritional profiles of spelt, emmer, einkorn, and kamut were presented. Reviewing data are valuable but in some cases the analysis is poor. Data presentation is sufficient, but their interpretation needs strengthening. However, I think that authors could add individual conclusions.
The authors should identify the main scope and the new knowledge of this research.
I have a few questions-observations. I believe that answering the questions will improve further the manuscript. Specific suggestions, comments, and questions are provided in the attached file.

Some editing for English language is required throughout the manuscript
Reviewer 2 Report
The work presented here is very interesting and well done, it is presented in a compact manner.
A small note: Tables are not transparent
Reviewer 3 Report
The authors reviewed the nutritional and sensory analysis of ancient wheats including spelt, emmer, einkorn, and kamut. The topic is interesting, well-organized, and well-written, which could attract the interest of Foods. However, some editing errors should be improved. I recommend minor revsion.
6 Nutritional and Health Benefits of Ancient Wheats
“Ancient wheat proteins are not suitable for manufacturing leavened baked products, but they do provide a different option for those who need to reduce their consumption of gluten overall. Ancient wheat proteins are not suitable for manufacturing leavened baked products, but they do provide a different option for those who need to reduce their consumption of gluten overall in addition to that ancient wheat grains have a rich chemical composition and a good quality of its baked products [21]” I think the sentence is repeated. Please check.
Please use Oxford comma, such as: a, b, and c, not a, b and c. Please check the whole manuscript.
Please provide full name of the abbreviations appeared in the manuscript for the first time.
6.2 “This subchapter delivers information on the most representative 9 macronutrients measured in ancient wheats”. I think it is inapropriate. Moisture and dry matter are not macronutrients. Please reconsider.
The symbol “-”should be changd as “–”, such as –18 °C, please check the whole manuscript.
Reviewer 4 Report
General
1. Provide a short history about ancient wheat including when and who cultivate it first.
2. Provide information about different types/species of ancient wheat.
3. How they were cultivated?
4. Does it different types still present and cultivated? If yes where are they cultivated?
5. Provide a map showing where ancient wheat was cultivated and where are now-a-days cultivated.
Introduction
1. Please rewrite the introduction by providing data on the topic from general to specific and at the end relate the introduction with your objectives of the study.
2. Provide a brief background about wheat taxonomy, ecology, soil type demand, climate etc.
3. Write the study objectives at the end of the introduction.
Conclusion
Please re-write it by including
1. Highlights what you have concluded from this study.
2. What you recommend for future?
3. What are the benefits of this study?
Minor revision for the language is needed.
Round 2
Reviewer 1 Report
Thank you for your effort. The manuscript was improved, and the authors responded satisfactorily in the most of my comments. However, authors are expected to answer our questions point-by-point, irrespective of whether they agree with the corrections-comments we have made. Ιn several cases the authors did not do so.
e.g page 5 “commodity grains’’ you mean commercial grains?- not answered
“Results of a study that was conducted in Poland measured the yielding parameters showed that ancient wheat species produced lower yields than modern wheat species” please add some details about sowing conditions, growing season and region.- by adding only the country where the experiment was carried out does not help to reach conclusions
Reviewer 4 Report
The authors have revised the manuscript according to the suggestions proposed.
Author Response
Thank you for considering the revisions of our manuscript.